# Building Flexible, Scalable, and Machine Learning-Ready Multimodal Oncology Datasets

**DOI:** 10.3390/s24051634

**Published:** 2024-03-02

**Authors:** Aakash Tripathi, Asim Waqas, Kavya Venkatesan, Yasin Yilmaz, Ghulam Rasool

**Affiliations:** 1Department of Machine Learning, Moffitt Cancer Center & Research Institute, Tampa, FL 33612, USA; asim.waqas@moffitt.org (A.W.); kavya.venkatesan@moffitt.org (K.V.); ghulam.rasool@moffitt.org (G.R.); 2Department of Electrical Engineering, University of South Florida, Tampa, FL 33620, USA; yasiny@usf.edu; 3Department of Neuro-Oncology, Moffitt Cancer Center & Research Institute, Tampa, FL 33612, USA; 4Department of Oncologic Sciences, University of South Florida, Tampa, FL 33612, USA

**Keywords:** cancer, oncology, multimodal, cloud computing, data lake, data warehouse, machine learning, deep learning, embeddings analysis

## Abstract

The advancements in data acquisition, storage, and processing techniques have resulted in the rapid growth of heterogeneous medical data. Integrating radiological scans, histopathology images, and molecular information with clinical data is essential for developing a holistic understanding of the disease and optimizing treatment. The need for integrating data from multiple sources is further pronounced in complex diseases such as cancer for enabling precision medicine and personalized treatments. This work proposes Multimodal Integration of Oncology Data System (MINDS)—a flexible, scalable, and cost-effective metadata framework for efficiently fusing disparate data from public sources such as the Cancer Research Data Commons (CRDC) into an interconnected, patient-centric framework. MINDS consolidates over 41,000 cases from across repositories while achieving a high compression ratio relative to the 3.78 PB source data size. It offers sub-5-s query response times for interactive exploration. MINDS offers an interface for exploring relationships across data types and building cohorts for developing large-scale multimodal machine learning models. By harmonizing multimodal data, MINDS aims to potentially empower researchers with greater analytical ability to uncover diagnostic and prognostic insights and enable evidence-based personalized care. MINDS tracks granular end-to-end data provenance, ensuring reproducibility and transparency. The cloud-native architecture of MINDS can handle exponential data growth in a secure, cost-optimized manner while ensuring substantial storage optimization, replication avoidance, and dynamic access capabilities. Auto-scaling, access controls, and other mechanisms guarantee pipelines’ scalability and security. MINDS overcomes the limitations of existing biomedical data silos via an interoperable metadata-driven approach that represents a pivotal step toward the future of oncology data integration.

## 1. Introduction

Clinicians routinely gather data from multiple sources to gain a deeper insight into patients’ health and provide tailored medical care. The reliance on multiple data sources for clinical decision-making makes medicine inherently multimodal, where the data modality refers to the form of data [1,2,3]. Each modality in such multimodal data may have a different resolution and scale due to its own data collection, recording, or generation process [4,5]. The data modalities may include (i) -omics information from genome, proteome, transcriptome, epigenome, and microbiome, (ii) radiological images from computed tomography (CT), positron emission tomography (PET), magnetic resonance imaging (MRI), ultrasound scanners or X-ray machines, (iii) digitized histopathology slides created using tissue samples and stored as gigapixel whole slide images (WSI), and (iv) electronic health record (EHR) that houses structured information consisting of demographic data, age, ethnicity, sex, race, smoking history, etc. and unstructured data such as discharge notes or medical reports [1,3,4].

Integrating data from heterogeneous modalities can create a unified, richer view of cancer, potentially more informational and complete than the individual modalities [1,6]. The multimodal medical data holds great potential to advance our understanding of complex diseases and help develop effective and tailored treatments [5,7]. The advent of high-throughput multi-omics technologies like next-generation sequencing (NGS), high-resolution radiological and histopathology imaging, and the rapid digitization of medical records has led to an explosion of diverse, multimodal data [8]. This data deluge has been a boon for machine learning, where abundant training data has directly enabled significant breakthroughs [9,10]. For example, the rise of large general-purpose datasets like Common Crawl for natural language processing (NLP) has fueled advances in language models and Artificial Intelligence (AI) assistants [11]. One may hope that extensive, standardized, and representative multimodal datasets in the medical domain would provide a fertile ground for developing advanced translational machine learning models. Machine learning thrives on massive, high-quality datasets; however, assembling such resources in healthcare poses unique challenges [12,13]. First, multimodal medical data is inherently heterogeneous and noisy, spanning structured (demographics, medications, billing codes), semi-structured (physician notes), and unstructured data (medical images). Aggregating such heterogeneous data requires extensive harmonization and manual processing. Second, reliability, robustness, and accuracy are critical for all medical applications [14,15,16]. However, real-world clinical data is often incomplete, sparse, and contains errors, which makes building robust and reliable models more challenging. Meticulous quality control and manual curation of these datasets are essential to train machine learning models [17,18]. Finally, strict data privacy and security considerations arise in healthcare, where data containd protected health information (PHI) that must be redacted, de-identified, and access controlled per the Health Insurance Portability and Accountability Act (HIPAA) [19,20].

Traditionally, vast amounts of multimodal data are generated during clinical trials and research studies where raw data undergoes initial processing and quality control by researchers. The data is then transmitted to standardization pipelines such as the National Cancer Institute’s (NCI) Center for Cancer Genomics (CCG) Genome Characterization pipeline [21], where the data is systematically annotated, formatted, and quality-controlled before being deposited into centralized biobanks. For example, NGS data from cancer genomic studies is standardized by CCG and deposited into the NCI’s Genomic Data Commons (GDC) [22]. However, imaging data from the same studies, consisting of CT, MRI, and PET scans, follow a different path and may end up in imaging archive like The Cancer Imaging Archive (TCIA) [23]. This leads to fragmentation of data across multiple disconnected databases. To address this, integrated data commons like the NCI Cancer Research Data Commons (CRDC) have been proposed [24]. The CRDC aims to link datasets from diverse sources using Findable, Accessible, Interoperable, and Reusable (FAIR) principles to enhance interoperability [25].

However, significant challenges remain in unifying multimodal data dispersed across repositories with heterogeneous interfaces, formats, and query systems. For example, a researcher studying lung cancer requires integrating clinical, imaging, and genomic data for their cohort across the GDC, TCIA, and other databases. But each has different application programming interfaces (APIs), schemas, and querying methods. Piecing together data manually across these silos is painstakingly difficult. There is a lack of unified interfaces and analytical tools that can work seamlessly across multiple cancer data repositories. This leads to isolated data silos and hampers easy access for multimodal data analysis. To address the limitations and fragmentation of current oncology data systems, we propose a novel solution called the “Multimodal Integration of Oncology Data System”, abbreviated as MINDS. MINDS is a scalable, cost-effective data lakehouse architecture that can consolidate dispersed multimodal datasets into a unified platform for streamlined analysis. To illustrate this, let’s consider the process of developing a machine-learning model using a limited dataset with and without the use of MINDS in Figure 1.

### Contributions of MINDS

MINDS makes several key objective contributions toward effectively managing and analyzing multimodal oncology data:1.Integrating siloed multimodal data into a unified access point: By consolidating dispersed datasets across repositories and modalities, MINDS delivers a single unified interface for accessing integrated data. This overcomes fragmentation across disconnected silos.2.Implementing robust data security and access control while supporting reproducibility: Strict access policies and controls safeguard sensitive data while enabling reproducibility via dataset versioning tied to cohort definitions.3.An automated system to accommodate new data continually: Automated pipelines ingest updates and additions, ensuring analysts always have access to the latest data.4.Enabling efficient, scalable multimodal machine learning: Cloud-based storage and compute scale elastically to handle growing data volumes while optimized warehousing delivers high-performance model training.

Apart from the above-mentioned achievements, MINDS has several novel aspects, including:The unprecedented scale of heterogeneous data consolidation enables new analysis paradigms. The cohort diversity in MINDS also surpasses existing systems.Tight integration between cohort definition and on-demand multimodal data assembly, not offered in current platforms.An industrial-strength cloud-native architecture delivers advanced translational informatics over a browser.Support for reproducibility via dataset versioning based on user cohort queries. This allows regenerating the same data even with newer updates.Option to build vector databases capturing data embeddings instead of actual data. This eliminates storage needs while ensuring patient privacy.

In this paper, Section 2 provides necessary background of the existing landscape of the multimodal heterogeneous datasets in oncology, from collection and processing to distribution. Section 3 delves into the methodology used to build the proposed data lakehouse architecture and discusses the project’s technical aspects in detail. In Section 4, we discuss the implementation results and the study’s potential implications on cancer research and clinics. Finally, Section 5 concludes with recommendations for future research.

## 2. Background and Literature Review

The rapid growth of biomedical data has created immense opportunities for translational research and significant data management challenges. Pioneering efforts have paved the way within this crucial domain by establishing needed infrastructure and principles over the past decades. These include caBIG [26] in 2004, interconnecting cancer researchers via an ambitious grid architecture, tranSMART [27] enabling customized cohort investigation, and i2b2 [28] spearheading flexible clinical data warehousing with temporal abstractions. However, as data scales intensify, core capabilities around scalability, provenance tracking, standardized metadata assimilation, and customizable cohort building, have created substantive yet addressable headroom for enhancements.

Emerging techniques like high-dimensional multimodal assay fusion [4,29] and multimodal data warehouses [30] have created new demands for consolidation platforms. By striving to synthesize the strengths of the seminal prior work while enhancing key dimensions like flexibility, replicability, and scalability, MINDS aims to stand on the shoulders of giants in pushing meaningful progress in addressing the constraints hampering reliable integrative modeling. Such demands motivate the development of new solutions to effectively consolidate, integrate, and analyze exponentially growing heterogeneous data types while accounting for the crucial lineage of achievements that collectively established the foundation. Below we discuss the existing methods of oncology data integration.

### 2.1. Data Characterization Pipeline

Standardized data characterization pipelines are vital in transforming raw biological samples into usable multimodal datasets. A sample data pipeline for gathering genomic modality from CCG for the GDC [22] is illustrated in Figure 2. The presented pipeline involves several stages, including tissue collection and processing, genome characterization, genomic data analysis, and data sharing and discovery. The NCI has adopted similar pipelines for medical images, referred to as the Imaging Data Commons [31] or IDC and Proteomics Data Commons or PDC [32].

Tissue Collection and Processing: Tissue source sites, which include clinical trials and community oncology groups, collect tumor tissue samples and normal tissue from participating patients. These samples are either formalin-fixed paraffin-embedded (FFPE) tissues or frozen tissue. In CCG, Biospecimen Core Resource (BCR) is responsible for collecting and processing these samples and collecting, harmonizing, and curating clinical data [21].Genome Characterization: This stage involves generating data from the collected samples. At CCG, the Genome Characterization Centers (GCCs) generate data from the samples received from the BCR. Each GCC supports distinct genomic or epigenomic pipelines, including whole genome sequencing, total RNA and microRNA sequencing, methylation arrays, and single-cell sequencing [21].Genomic Data Analysis: The raw data from the previous stage is then transformed into meaningful biological information at this stage. In CCG, the Genomic Data Analysis Network (GDAN) transforms the raw data output from the GCCs into biological insights. The GDAN has a wide range of expertise, from identifying genomic abnormalities to integrating and visualizing multi-omics data [21].Data Sharing and Discovery: At this stage, the insightful genomic data is processed, shared, and unified at a central location. The NCI’s Genomic Data Commons (GDC) harmonizes genomic data by applying a standardized set of data processing protocols and bioinformatic pipelines. The data generated by the Genome Characterization Pipeline are made available to the public via the GDC [21,22].

### 2.2. Traditional Data Management—BioBanks

Traditionally, medical data modalities are stored and managed separately in biobanks. These biobanks are the repositories that store biological samples for use in research and by clinicians for reference. Today, such biobanks have become an essential resource in medical and oncological facilities [33]. They provide researchers access to various medical samples and associated clinical and demographic data, which is used to study disease progression, identify biomarkers, and develop personalized and new treatments. However, traditional data management using biobanks has several limitations, enumerated below:Fragmented Data: One of the main issues is that data from different sources are often stored in separate biobanks, leading to fragmentation of information [34]. This makes integrating and analyzing data across different modalities difficult, limiting the potential for comprehensive, multi-dimensional analysis of patient data [33].Incoherent Data Management: How data is stored, formatted, and organized often varies significantly across biobanks, even for the same patient. For example, clinical data may be encoded differently, imaging data may use proprietary formats, and terminology can differ across systems. This heterogeneity and lack of unified standards make aggregating and analyzing data across multiple biobanks challenging [33].Data Synchronization: Over time, patient data stored in separate biobanks tends to go out of sync as patients undergo new tests and treatments, adding new data to different silos uncoordinatedly [33]. Piecing together a patient’s history timeline requires extensive manual effort to sync disparate records across systems [33].Data Governance: The increasing prevalence of bio-banking has sparked an extensive discussion regarding the ethical, legal, and social implications (ELSI) of utilizing vast quantities of human biological samples and their associated personal data [35]. Ensuring and safeguarding the fundamental ethical and legal principles concerning research involving human data in Biobanks becomes significantly more intricate and challenging than conducting ethical reviews for specific research projects [35].

### 2.3. Data Commons

The concept of data commons has emerged to address the challenges faced by biobanks. A data commons is a shared virtual space where researchers can work with and use data from multiple sources. The NCI has developed the CRDC, which integrates different data types, including genomic, proteomic, imaging, and clinical data, into a unified, accessible platform [24]. The CRDC provides researchers access to various data repositories, including the GDC, PDC, and IDC. Each of these repositories hosts a specific data type, and together, they form a comprehensive platform for multimodal data analysis. While the CRDC has made significant strides in integrating diverse data types, it still faces challenges. One of the main issues is the difficulty in harmonizing data from different sources. Due to the differences in data formats, standards, and quality control measures across data sites and modalities, it takes significant effort by the researchers to conform the data to uniform quality standards. The Cancer Data Aggregator (CDA) was developed to address this issue and facilitate data integration across different data commons. CDA provides an aggregated search interface across major NCI repositories, including the PDC, GDC, and IDC. It allows unified querying of core entities like subjects, research participants, specimens, files, mutations, diagnoses, and treatments, facilitating access across different data types [36]. CDA has limitations, like static outdated mapping and the inability to incorporate external repositories. This motivates the need for more robust integrative platforms. The proposed MINDS system aims to overcome these challenges in several key ways:CDA’s mapping of the CRDC data is not real-time. For example, as of September of 2023, when querying patients with the primary diagnosis site being lung, only 4870 cases are present, despite there being 12,267 cases present in the GDC data portal. MINDS pulls source data directly from repositories like GDC to ensure real-time, up-to-date mapping of all cases.MINDS is designed as an end-to-end platform for users to build integrated multimodal datasets themselves rather than a fixed service. The open methodology enables full replication of huge multi-source datasets. To this end, anyone can replicate our method to generate the exact copy of over 40,000 public case data on their infrastructure.MINDS is flexible and incorporates diverse repositories and data sources, not just CRDC resources. Our proposed architecture can integrate new repositories as needed, unlike CDA, which is constrained to CRDC-managed data. For example, the cBioPortal for Cancer Genomics, a widely used platform for exploring, visualizing, and analyzing cancer genomics data, has its own data management and storage system separate from the CDA [37,38]. The data stored in cBioPortal cannot be directly queried or accessed through CDA, limiting the potential for integrated data analysis across platforms.

### 2.4. The “Big Data” Approach

We have used the Big Data approach in our work [12,13]. Among the recent advancements in healthcare data management, the big-data approach is the most prominent and feasible solution [7,8,39]. The rapid technological progress has led to an unparalleled utilization of computer networks, multimedia, internet of things, social media, and cloud computing, resulting in an overwhelming generation of “big data” [40]. Effectively collecting, managing, and analyzing vast amount of healthcare data through big data processing has become crucial. The big data processing involves various techniques, such as data mining, leveraging data management, machine learning, high-performance computing, statistics, and pattern recognition to extract knowledge from extensive datasets. These datasets possess distinctive characteristics, often called the seven *V*s of big data, as shown in Figure 3. The Big Data approach guides data handling strategies. By considering each of these aspects, we can effectively manage oncology data and, in turn, build better, effective models. We use two primary data management systems to facilitate our big data approach: Data Warehouses and Data Lakes.

Data warehouses represent a foundational pillar of the big data paradigm. A data warehouse integrates heterogeneous data from diverse sources into a centralized, well-organized repository to enable analysis. This repository provides a highly structured environment explicitly optimized for analytics, reporting, and deriving insights across vast information [40]. By fulfilling this role, data warehouses deliver immense value in informed decision-making. The process of assembling data into warehouses is called data warehousing. “Schema-on-write” is the core concept employed, where the warehouse schema is predefined to meet specific analytical needs before data is loaded. This upfront structural optimization makes warehouses ideal for handling structured data. Supervised machine learning thrives in warehouses, as structured, consistent data facilitates training algorithmic models. Moreover, the innate high degree of organization enables fast, efficient querying to uncover trends and patterns through predictive analytics [40]. Overall, by structuring varied data sources into a unified environment purpose-built for analytics, data warehouses provide the backbone for deriving value from big data across many domains.

Data lakes complement the warehouses by providing centralized but low-structure storage to accumulate expansive, heterogeneous data in raw form. In contrast to “schema-on-write”, data lakes employ “schema-on-read”, which only defines structure when data is queried. This provides flexibility to modify analytics on-demand [40]. With their innate tolerance for storing original, unprocessed data, lakes accommodate structured, semi-structured, and unstructured data types. The lack of enforced structure enables rapid scaling to meet growing analytics demands. The dual architectures of data warehouses and data lakes provide structured refinement and raw accommodating capabilities to put big data into action. Lakes aggregate heterogeneous datasets, while warehouses prepare refined data for analysis. This symbiotic combination ultimately enables MINDS to derive maximal value from oncology’s multidimensional data landscape.

### 2.5. Summary of Gaps in Existing Methods

While prior work has laid crucial foundations, several persistent constraints around consolidation, interoperability, scalability, provenance, and security have encumbered reliable integrative modeling on multimodal data. Biobanks carry siloed modalities with heterogeneous formats, creating barriers to unification and requiring extensive manual synchronization effort. Data commons combined various data types into unified platforms but lack harmonization of diverse data sources. Static mappings fail to reflect repositories’ real-time state, while disjoint querying systems limit holistic analysis across databases. Fundamentally, past efforts centered on aggregating principally structured sources, lacking the breadth to effectively harness the heterogeneity spanning images, assays, text, and sensors. With data volumes intensifying across these manifold streams, inflexible on-premises systems strain to provide needed scalability. Reproducibility suffers from dynamic dataset derivation as model provenance linkages fade. Finally, while ethical rigor grows in importance with scale, most architectures offer worryingly coarse-grained control over access policies. By tackling this multiplicity of the persistent challenges through enhancements leveraging the prior seminal achievements, MINDS aims to advance reliable, responsible multimodal modeling on big oncology data. The key limitations that constrain multimodal integrative modeling through existing approaches are summarized below:Prior consolidation is limited to structured data: Most prior efforts, like CDA, focused on consolidating structured clinical records. Support for aggregating unstructured imaging, -omics, and pathology data is lacking.Query interfaces have limited standardization: Different repositories have proprietary APIs and schemas. Unified interfaces for federated querying are needed.Scalability is constrained for large data: On-premises systems restrict scaling storage and compute for exponentially growing heterogeneous data.Minimal reproducibility without versioning: Dynamic dataset extracts make precise tracking of model data versions difficult, hampering reproducibility.Coarse-grained access controls: Most systems have limited options for fine-grained data access policies tailored to users.

Addressing these gaps is pivotal to unlocking translational applications of multimodal oncology data through enhanced consolidation, standardization, scalability, provenance, and security. By tackling each limitation, MINDS aims to overcome persistent bottlenecks that have hitherto encumbered reliable integrative modeling on heterogeneous big data.

## 3. Methodology

This section details the technical implementation of the proposed MINDS architecture. We begin by presenting the high-level requirements that informed key architectural decisions. We then dive into the three-stage architecture of MINDS, describing each component and its role in enabling scalable and secure management of multimodal oncology data. Next, we provide deployment options for MINDS, including details on implementing the system in the cloud across diverse platforms and on-premise infrastructure. Finally, we outline key use cases and user interactions with MINDS.

### 3.1. Requirements of a Flexible and Scalable Data Management System

To handle the complexities, scales, and heterogeneity in the structure and function of oncology data, the data management system design has to be comprehensive, scalable, and interoperable. The primary goal of this system is to cater to the needs of machine learning engineering, which requires a robust and efficient data management infrastructure to build accurate and reliable models. We set off with the aim to design and build a data management system with the following requirements in mind:Requirement 1: Minimize large-scale unstructured data storage whenever possible. This requirement ensures the efficient use of storage resources and allows the user to access the data directly from the data provider.Requirement 2: The system should be horizontally and vertically scalable. Satisfying this requirement is crucial to handle the increasing volume of oncology data and ensure the system can accommodate data size and complexity growth.Requirement 3: The system should be interoperable, allowing for the easy integration of new data sources. This is important in oncology, where data is often distributed across various databases and systems.Requirement 4: The system should track data from the point of ingestion to the point of training. This ensures reproducibility, a key requirement in scientific research and machine learning.Requirement 5: Incorporate audit checkpoints in the data collection, pre-processing, storage, processing, and analysis stages of the data pipeline. This ensures data integrity, the prime consideration in delivering reliable machine learning outcomes.

### 3.2. MINDS Architecture

Considering the above-mentioned requirements, we have built a Multimodal Integration of the Oncology Data System (MINDS). The MINDS system design adopts a common two-tier data architecture, a data lake, and a data warehouse [40] to process data and derive meaningful insights efficiently. Figure 4 illustrates the architecture of MINDS, which is divided into three primary stages: (1) Data Acquisition, (2) Data Processing, and (3) Data Serving. Key goals include scalability of individual components, interoperability via standardized APIs and schemas, security leveraging authentication and encryption, and usability across interactive and programmatic access patterns. To meet this requirement MINDS is built using the cloud-based technology of Amazon Web Services (AWS), the cloud-based architecture allows us to scale up or down easily based on the data volume requirements and the required computational resources. It also provides a wide range of tools and services that can be leveraged to build, deploy, and manage a data management system. While the current MINDS implementation leverages AWS, the architecture is designed to enable deployment across different cloud platforms, not just AWS. The core methodology centers on interfacing with managed cloud services, abstracting the underlying infrastructure through common programmatic interfaces. This service-oriented approach enhances portability and avoids extensive customization tied to a single provider. For example, the S3 storage layer could be replaced with Google Cloud Storage buckets, AWS Glue with Azure Data Factory, RDS and Redshift with Snowflake’s data platform, and Lambda with Cloud Functions. The overall system architecture would remain consistent while swapping the provider services. When migrating platforms, trade-offs exist around performance, access controls, and other factors. But by using managed services with standard APIs, MINDS aims for platform-independent portability. Figure 5 provides a detailed layout of technical components at each stage using AWS cloud infrastructure and the tools utilized to actualize the system. Definitions of these technical components are summarized in Box 1.
Box 1.Definitions of key cloud components.Amazon S3 Ingest BucketObject storage bucket for staging raw data before loading into a data lake.Amazon Web Services (AWS)A cloud platform that provides scalable computing, storage, analytics, and machine learning services.AWS AthenaServerless interactive query service to analyze data in Amazon S3 using standard SQL.AWS Big Data AnalyticsSuite of services for processing and analyzing big data across storage, compute, and databases.AWS Data Lake FormationService to set up and manage data lakes with indexing, security, and data governance.AWS Data WarehouseFully-managed data warehousing service for analytics using standard SQL.AWS Glue CrawlerDiscovers data via classifiers and populates the AWS Glue Data Catalog.AWS Glue Data CatalogCentral metadata store on AWS for datasets, schemas, and mappings.AWS LambdaServerless compute to run code without managing infrastructure.AWS QuickSightBusiness intelligence service for easy visualizations and dashboards.AWS RDSAmazon Relational Database Service is a managed relational database service that handles database administration tasks like backup, patching, failure detection, and recovery. Including RDS MySQL, a managed relational database optimized for online transaction processing.AWS RedshiftPetabyte-scale data warehouse for analytics and business intelligence.JDBCJDBC (Java Database Connectivity) is a standard API for connecting to traditional relational databases from Java. The JDBC was released as part of the Java Development Kit (JDK) in version 1.1 in 1997 and has since been part of every Java edition.


#### 3.2.1. Stage-1: Data Acquisition

Data sources: Data acquisition is the first and crucial step in building the MINDS platform. This process involves gathering all publicly available structured and semi-structured data from the data sources. As mentioned earlier, the CRDC and other oncology data management initiatives host vast amounts of patient information, and we use them as the primary data sources for our system. These sources primarily include the three data commons portals, GDC, IDC, and PDC. Additionally, we use the CRDC’s Cancer Data Aggregator (CDA) tool to map all the patient information across the commons into one cohesive database. This database then expands to accommodate the patient data stored across other portals, such as the cBioPortal, Xena, and other relevant data sources [37,38,41]. It is pertinent to mention that we do not store any unstructured data in MINDS, such as whole slide images or radiology scans. MINDS instead pulls the unstructured data from their respective data commons based on the cohort the users want to build and the modalities they require for processing through the portal APIs. Hence, we are not required to store large unstructured data such as gigabyte pathology images in our database.

For the initial version of MINDS, we leverage the GDC as the primary data source due to its comprehensive collection of up-to-date, publicly available oncology data. The GDC portal contains clinical, biospecimen, and molecular data across diverse cancer studies, representing over 86,000 cases spanning 78 projects. The GDC has the most extensive public data holdings out of the three NCI data commons. As of 2023, it hosts over 3 petabytes of genomic and clinical data from the NCI programs like The Cancer Genome Atlas (TCGA) and Therapeutically Applicable Research to Generate Effective Treatments (TARGET). The GDC also has a well-designed and detailed data model that structures and connects the clinical, biospecimen, and molecular data domains. The availability of this robust data dictionary and schema metadata makes the ingestion and integration of new GDC datasets simpler and more consistent. Leveraging thousands of richly annotated multi-omic cancer profiles, we can develop integrative and predictive models by utilizing all the public cases in the GDC for MINDS initial deployment. The breadth of tumor types enables the building of generalized models applicable across different cancers. As the MINDS data repository expands to incorporate more primary sources beyond GDC, the experience of integrating the GDC data provides a solid foundation to build upon. The tooling ETL workflows developed to ingest and harmonize GDC data can be extended to transform and connect new oncology datasets into the MINDS knowledge system.

#### 3.2.2. Stage 2: Data Processing

A foundation of the MINDS architecture is ingesting petabytes of structured clinical, biospecimen, and molecular data from cancer genomic repositories like the GDC. This raw metadata arrives in heterogeneous formats including JSON documents, CSV exports, and XML messages conveying case details, lab assays, pathology reports, and tissue sample attributes. While information-rich, effectively using this disjointed data to drive integrative insights requires significant wrangling. We leverage the GDC common data model as an integration schema to streamline aggregation and analysis. This model structures entities like cases, files, and read groups into a normalized graph representation, with nodes denoting key objects and edges linking related records. For example, a case entity may reference constituent pathology reports, sequencing files, or tissue aliquots to provide a unified view spanning this network of connected data. The GDC data dictionary rigorously defines properties and relationships to provide semantic consistency. Aligning raw data to this canonical representation enables unified storage, queries, and computational pipelines. However, the raw downloads natively arrive in varied shapes. JSON clinical documents describe patient characteristics differently than TSV biospecimen exports. Our first challenge is flexible parsing and mapping.

Adopting Interoperability Standards: The need to integrate data from multiple sources is further pronounced in complex diseases such as cancer when considering efforts such as precision medicine and personalized treatments. However, interoperability remains a major challenge in practice despite extensive standards development. Many clinical, genomic, imaging, and literature databases use disjoint interfaces, formats, and terminologies, thus hampering unified analytics. Several domain-agnostic standards have emerged to promote harmonization:Health Level 7 (HL7): Defines structures and semantics for messaging healthcare data between computer systems, including Clinical Document Architecture and Fast Healthcare Interoperability Resources (FHIR) specifications [42,43].Fast Healthcare Interoperability Resources (FHIR): Specifies RESTful APIs, schemas, profiles, and formats for exchanging clinical, genomic, imaging, and other healthcare data. Offers platform-agnostic interconnection [43].Clinical Data Interchange Standards Consortium (CDISC): Develops data models, terminologies, and protocols focused specifically on clinical research and FDA submissions, including the Study Data Tabulation Model (SDTM) and the Clinical Data Acquisition Standards Harmonization (CDASH) [44].

However, adopting these standards remains inconsistent, and significant translator development is required to bridge entities [45]. The tight coupling of databases to proprietary representations threatens interoperability. Furthermore, medical ontologies and terminologies such as those given below play a crucial role in promoting both human and machine-readable shared understanding:Systematized Nomenclature of MEDicine Clinical Terms (SNOMED CT): Provides consistent clinical terminology and codes for electronic health records. Enables semantic interoperability [46].National Cancer Institute (NCI) Thesaurus: Models cancer research domain semantics with 33 distinct hierarchies and 54,000 classes/properties. Binds related concepts for knowledge discovery [47].

The GDC data model and dictionary enhance interoperability by structuring and defining entities, properties, and relationships standardized. When ingesting data, the AWS Glue crawler leverages these common semantics to map input elements into the unified representation. This semantic alignment enables integrated analysis despite originating heterogeneity.

Data Dictionary, Schema, and Entity Relationships: GDC structures clinical, biospecimen, and molecular data using a consolidated data model that interconnects related entities into a directed acyclic graph (DAG) representation. This data model underpins the organization and semantics of the petabyte-scale GDC dataset. The model comprises a network of nodes representing key data objects (cases, samples, reads, etc.) linked through edges denoting relationships. Nodes have properties like type, age, and tumor_stage, while edges characterize affiliations like a sample derived_from a case. Robust semantic definitions specify permitted nodes, their properties, associated data types, and linkage rules. This ontology ensures consistency critical for downstream interoperability. Technically, the data model utilizes a mix of JSON and YAML schemas coupled with Python 3 and SQL codebases to architect domain representations. Schemas define valid elements and constraints serialized into JSON documents. Codebases ingest and query documents while preserving compliance. The GDC dictionary elaborates on metadata driving consistency. For example, the sample entity has documented required fields like sample_type and permissible values like Solid Tissue. The authentication service verifies submitted entities to satisfy specs. At the infrastructure layer, dictionaries transform into SQLite representations. Indexed tables track datasets while optimized queries fetch connections. Although decentralized, federated services coalesce distributed systems into an integrated data collaboration. The presentation tier visualizes linkages via an entity-relationship diagram highlighting cardinality rules (one-to-many mappings, etc.). Users traverse graphs accessing constituent records through REST APIs. By providing rigorous blueprints governing content packaging, exchange, and interpretation, GDC data models power external consistency unlocking unified workflows spanning partners—enabling interconnected explorations.

#### 3.2.3. Stage 3: Data Serving

We provide two core methods for researchers to consume processed oncology data based on workflows. We built interactive dashboards for interactive visualization and cohort analysis that use the data stored in the warehouse. Additionally, for developers and computational researchers needing to ingest data into pipelines, we provide an open-source Python toolkit as part of MINDS that programmatically downloads the unstructured multimodal data from disparate databases.

Dashboard: At the data consumption stage, the structured data in the data warehouse is utilized for various purposes. The data consumption process is designed to provide users with an interactive and intuitive interface for exploring, visualizing, and analyzing the data. This is achieved through a dashboard built on Amazon QuickSight [48], a fully managed business intelligence service that enables data visualization and interactive analysis. Users can interact with the dashboard to explore various aspects of the data and identify trends, patterns, or correlations using QuickSight’s machine learning-powered insights.

Figure 6 presents sample visualizations enabled by the MINDS analytics dashboard, allowing researchers to explore different data attributes like the cause of death and tumor subtype distributions. For example, the death date graph reveals a peculiar underreporting anomaly between 2014–2017 that may warrant investigation into potential data quality issues. Meanwhile, tumor classification breakdown identifies pancreatic cancer as the most represented diagnosis, informing potential studies targeting prevalent categories.

Beyond distributions, the interactive dashboards may also catalyze discoveries by empowering explorations into relationships between clinical factors, assays, and outcomes. As illustrated, researchers could assess survival trends across cancer subtypes to uncover prognostic biases. Recurrence patterns may be analyzed with modalities like genetic mutations and treatment regimens to reveal predictive biomarkers or personalized medicine insights. Apart from the analytical categories depicted in Figure 6, the MINDS analytics dashboard allows the researchers to filter data based on any clinical or biological fields such as age, gender, ethnicity, tumor grade, treatment type, year of diagnosis, survival, etc.

Unstructured Data Download Tools: MINDS enables users to build focused, multimodal datasets for targeted analysis by combining warehouse-driven cohort queries with automated unstructured data collection. Patient cohorts are defined by querying the database directly through SQL. The case IDs can be extracted from the cohort, and the resulting list of case IDs is used to retrieve all related unstructured data from the GDC, IDC, and PDC portals using their respective API interfaces. As part of the MINDS toolkit, we provide a Python utility that accepts the case ID list as input and programmatically calls the APIs to bulk download images, pathology, -omics, and other files for those specific cases. The downloaded data is organized into a folder structure with a top-level “/raw” folder containing subfolders for each case ID. Each case folder contains the unstructured data objects from GDC, IDC, and PDC for that case. JSON manifest files are also generated to capture metadata like file IDs, types, and sources. This enables easy indexing and querying of the unstructured data extracts.

### 3.3. Cloud Deployment

This section outlines the AWS cloud implementation of MINDS, leveraging core infrastructure services to enable scalable data aggregation, processing, and unified access. Our approach incorporated several key big data techniques essential to the MINDS architecture. We utilized Amazon S3 for distributed storage, creating a data lake environment capable of handling petabytes of heterogeneous data. AWS Redshift and EMR were employed for large-scale data warehousing and distributed SQL and Spark processing, respectively. These services enabled the building of high-performance SQL query engines and the efficient processing of large data volumes. AWS Glue played a critical role in machine learning-powered ETL, allowing for the transformation and structuring of data for analysis. Serverless computing using AWS Lambda was instrumental in managing scalable workloads, preventing server overflow, and ensuring system responsiveness. Together, these components formed a robust foundation essential for addressing the challenges of volume, variety, and velocity inherent in the multimodal oncology data within MINDS. While the current system deployment utilizes Amazon Web Services, the underlying architecture is designed for portability across cloud platforms. By interfacing with common storage, database, analytics, and machine learning modules rather than low-level servers or virtual machines, much of the MINDS technology stack can be replicated on alternate providers.

#### 3.3.1. MINDS Infrastructure on AWS

Data Acquisition Process: We pull all semi-structured and structured data from the GDC data portal for all public cases, including TSV and JSON files containing various clinical (clinical, exposure, family history, follow-up, and pathology detail) and metadata of biospecimen (aliquot, analyte, portion, sample, and slide) information. This data is then uploaded into an Amazon S3 Ingest Bucket [49]. This bucket acts as the staging storage for the data before it is uploaded to the data lake. To orchestrate the full data lake setup, we utilize the AWS Data Lake Formation tool [50], which automates the transformation of the semi-structured data stored in the S3 bucket into a queryable data lake using AWS Glue crawlers to catalog the data and store it in data tables [51].

Data Updating Process: The data acquisition is not a one-time event but a continuous process that must be updated regularly to ensure the data lake is always up-to-date with the latest data. The new data is not uploaded arbitrarily but rather arrives through scheduled ETL routines that run every 12 h to poll source repositories like GDC using their APIs. For example, scripts leverage the GDC REST API to query for newly added cases, files or metadata since the last update based on a timestamp. The incremental changes are downloaded via the API and uploaded to the S3 bucket on a Linux-based cron schedule, such as daily at 9 a.m. UTC. This polling pattern is tailored for each integrated data source and its API capabilities. Explicitly tracking data provenance through structured ingestion and ETL ensures the S3 bucket receives only authorized data uploads, avoiding random additions. We use AWS Lambda serverless compute [52] to trigger Glue crawlers automatically whenever new data lands in the S3 bucket. This ensures our data lake is always up-to-date with the latest data without explicit manual synchronization. This also helps reduce the data transfer rates because the system updates the data lake only with the delta between the bucket and the data lake. The data acquisition process is designed to be robust and scalable, capable of handling the increasing volume of oncology data. It also ensures the safety and integrity of the data by establishing secure connections to the databases from which data needs to be extracted.

Data Extraction and Transformation to Structured Format: Once the data is acquired, the next step is to clean, process, and aggregate this data. At this stage, the data is extracted from the data lake, transformed into a more structured format, and loaded into the data warehouse. This is done using Amazon AWS Glue 4.0 [53], which ensures consistency and compatibility across data types and sources. AWS Glue performs the ETL actions using the AWS Glue crawler [51]. The crawler works in a series of steps to ensure the data is appropriately cataloged and ready for analysis. Figure 7 shows the internal workings of the AWS crawler that ensure the data is properly processed and ready for analysis, making it easier for users to extract valuable insights from the data. The steps involved in the AWS crawler workflow are as follows:1.Establish access-controlled database connections: The crawler first establishes secure connections to the databases from which data needs to be extracted. This ensures the safety and integrity of the data in transit.2.Use custom classifiers: If any custom classifiers are defined, they catalog the data lake and generate the necessary metadata. These classifiers help in identifying the type and structure of the data.3.Use built-in classifiers for ETL: AWS’s built-in classifiers perform ETL tasks for the rest of the data. This process involves extracting data from the source, transforming it into a more suitable format, and loading it into the data warehouse.4.Merge catalog tables into a database: The catalog tables created from the previous steps are merged into a single database. During this process, any conflicts in the data are resolved to ensure consistency and deduplication.5.Upload catalog to a data store: Finally, the catalog is uploaded to a data store to be accessed and utilized for analytics. This data store is a central repository where all the processed and cataloged data is stored.

When ingesting data, the AWS Glue crawler parses source elements into this consolidated model by mapping input fields into the GDC dictionary. For instance, a Read Group JSON would have its metadata properties (like ID, library name, etc.) inserted as columns into the standardized Read Group table definition used across MINDS while retaining references to the parent Case/File IDs to recreate linkages. The unified representation enables joining and analysis across interconnected data domains related to biospecimen, sequencing, diagnoses, etc., even if originating formats vary. This ensures interoperability among diverse data sources through a common but fast health interoperable resource. To incorporate emerging repositories into this existing data model, we extract salient clinical and experimental metadata based on publication schemas and use the flexible AWS Glue schema evolution tools to extend existing definitions or spawn new tables aligned with import sources if needed. Templatized mapping configurations adjust for input heterogeneity while producing consistent MINDS representations to power integrated SQL queries across past and future data partners - avoiding isolated silos or reengineering efforts when onboarding additional cohorts. Hence, MINDS has built-in scalability supported by interoperable functions. The crawler uses the GDC node schema definitions in YAML files to parse the JSON documents and infer the schema. The GDC case entity is defined with properties like case_id, disease_type, demographic, diagnoses, etc. When the crawler processes a case JSON document from the GDC portal, it maps the JSON properties to columns in a Glue table definition based on the GDC data model. This way, the GDC model’s underlying graph structure transforms relationships into a relational view. The Glue crawler output is a table definition in the AWS Glue Data Catalog. Users can directly query and join with other clinical, biospecimen, and genomic tables ingested from GDC. The dictionaries also provide metadata like each property’s data types and code lists. When creating data definition language (DDL) for the tables, the crawler leverages this to assign appropriate column types, formats, and validations. This helps maintain data integrity and consistency during the transformation process.

Uploading Data to Warehouse: The normalized clinical, biospecimen, and molecular data cataloged by the AWS Glue crawler undergoes loading into Amazon Redshift, which serves as the primary data warehouse for enabling high-performance analytics. With this structured data, we also populate an Amazon RDS MySQL cluster to support efficient inserts and updates as new data arrives from source systems. However, given its optimization for such read-heavy workloads, analytical queries are routed directly to Redshift [54]. As a petabyte-scale massively parallel processing (MPP) data warehouse service, Amazon Redshift employs advanced query processing, adaptive machine learning optimizers, and columnar storage layouts purpose-built for complex aggregations, filters, and joins across huge datasets. By leveraging separate data warehouse and transactional database environments, MINDS supports fluid exploration without impacting critical path operations that rely on consistent low-latency database performance unaffected by ad hoc analysis. We incrementally load the Glue-cataloged data into Redshift using high-throughput COPY commands to enable fast bulk data movement from S3 object storage. Redshift Spectrum interfaces create external tables pointing directly at structured datasets in S3 buckets, providing direct access without loading the data into local warehouse storage. This allows interactive SQL analytics directly on raw JSON, CSV, and TSV objects with automatic inferencing of schema and transformations to perform as data is read at query runtime. The centralized AWS Glue Data Catalog manages table definitions, schemas, partitions, and mappings across these disparate storage and processing environments—serving as the primary metadata store and enabling unified access to explore and visualize data across tools like Amazon QuickSight, Amazon Athena [55], and Amazon SageMaker. We leverage Athena’s serverless SQL query engine to enable users to analyze the consolidated data using standard ANSI SQL without needing to connect to the underlying data stores, enhancing accessibility directly.

#### 3.3.2. Benefits of Cloud as a PaaS Platform

In Platform as a Service (PaaS), the cloud’s intrinsic security features are not just an add-on; they form the bedrock of a comprehensive data protection strategy. MINDS utilizes the built-in security of cloud platforms to protect data. We implement several security services from AWS to ensure our data storage and processing are safe and private. This includes security, management, and backup mechanisms.

Security in S3 and Data Lake: Security and management are critical aspects of any data management system, especially when dealing with sensitive medical data. In MINDS, we employ several AWS security services and best practices to ensure the highest data security and privacy level. Amazon S3, where our data lake resides, provides robust security capabilities. We have enhanced these with network traffic encryption using TLS 1.2 and enforcing data integrity with HTTPS. All data in S3 is encrypted at rest using 256-bit Advanced Encryption Standard (AES) keys managed through AWS Key Management Service (KMS). Additionally, we use Identity and Access Management (IAM) policies to precisely manage access at both the resource and action levels, utilizing temporary credential chains to avoid exposure to raw secrets. Our setup includes Virtual Private Cloud (VPC) endpoints to prevent public exposure of the data.

Security in Data Warehouse: Our data warehouse, Amazon Redshift [54], incorporates multiple layers of security to protect sensitive oncology data. It integrates with AWS IAM, allowing fine-grained access control to resources. Data in transit to and from Redshift is protected using SSL connections. For data at rest, Redshift employs encryption using Key Management Service (KMS) and Hardware Security Module (HSM) encryption for large volumes exceeding terabytes. Redshift also enforces strict SQL-based authorization to ensure secure data access [56]. Furthermore, we utilize features like Virtual Private Cloud (VPC) for network isolation and comprehensive audit logging and compliance certifications for enhanced security and accountability.

Security in ETL and Dashboard: We adhere to stringent security practices in the context of data processing and ETL with AWS Glue. AWS Glue is integrated with AWS Lake Formation, which allows for fine-grained, column-level access control, ensuring that only authorized personnel can access sensitive data. AWS Glue ETL jobs run in a secure and isolated environment, with all necessary resources provided by AWS Glue [57]. This is complemented by regular updates to server security groups, operating system patches, and adherence to the Center for Internet Security (CIS) hardening guidelines. For data consumption, Amazon QuickSight employs AWS IAM and AWS Lake Formation for robust access control, supporting both encryption at rest via AWS KMS and encryption in transit using SSL. Additionally, AWS CloudTrail provides detailed audit logs, enabling effective incident investigation and response.

Monitoring and Audit Logging: In addition to the above-mentioned security measures, we also employ monitoring and logging using AWS CloudTrail and Amazon CloudWatch [58]. These services provide visibility into user activity and API usage, allowing us to detect unusual or unauthorized activities. This helps build audit trails and trigger security events in case of an undesired action. We also use Amazon RDS Multi-AZ deployments for redundancy, high availability, and failover support for database instances. Multi-AZ creates a primary RDS instance with a synchronous secondary standby instance in another Availability Zone (AZ) for enhanced redundancy and faster failover.

Backups and Recovery Mechanisms: MINDS leverages AWS services’ robust backup, redundancy, and disaster recovery capabilities to maximize system availability and protect against data loss. Amazon S3 buckets are versioned, with all object modifications saved as new versions. This allows restoring to any previous version. Cross-region replication sends object replicas to geographically distant regions to mitigate region-level failures. S3 object lock prevents accidental deletions during a specified retention period. RDS clusters run as Multi-AZ deployments with a standby replica in a secondary AZ for high availability, automatic failover, and fast recovery. Point-in-time restore rolls back to previous database states using retained backups. Database snapshots are stored in S3 for long-term durability. Redshift distributes replicas across nodes for local redundancy. It replicates snapshots and transaction logs to S3 to protect against node failures. Snapshots can restore clusters to any point in time. Combining versioning, redundancy, failover capabilities, and recovery automation, MINDS provides resilience against failures and minimizes disruption. Robust security protects against data loss from malicious events.

#### 3.3.3. Scalability across Different Platforms

While the current MINDS implementation leverages AWS, the architecture is designed to enable deployment across different cloud platforms, not just AWS. The core methodology centers on interfacing with managed cloud services, abstracting the underlying infrastructure through common programmatic interfaces. This service-oriented approach enhances portability and avoids extensive customization tied to a single provider. For example, as shown in Figure 8, the S3 storage layer could be replaced with Google Cloud Storage buckets, AWS Glue with Azure Data Factory, RDS and Redshift with Snowflake’s data platform, and Lambda with Cloud Functions. The overall system architecture would remain consistent while swapping the provider services. When migrating platforms, trade-offs exist around performance, access controls, and other factors. But by using managed services with standard APIs, MINDS aims for platform-independent portability. The MINDS architecture can be replicated to the Google Cloud Platform to demonstrate feasibility through the following replacement and compatibilities.

Employing Cloud Data Fusion for data integration in place of AWS GlueLeveraging BigQuery for data warehousing rather than RedshiftUsing Cloud SQL over RDS for relational dataAdopting Cloud Functions and Cloud Run for serverless compute instead of Lambda.Visualizing with Looker as an alternative to QuickSightApplying Cloud Data Loss Prevention for security rather than AWS options

### 3.4. On-Premise Deployment

While the cloud delivery model provides advantages like elastic scalability, hands-off management, and usage-based costing, some organizations may prefer on-premise deployment of MINDS due to data sovereignty, customization, or latency constraints. Despite extensive security protections, regulated data may mandate localized processing. Custom modules like augmented analytics dashboards may also require internal hosting. We provide an open-source Python toolkit for configurable local installations to address these needs while retaining MINDS’ consistent methodology.

The MINDS library abstracts the orchestration of storage, databases, and web services into simple commands. A Docker container runs the setup scripts to bootstrap a production-ready environment. This generates a MySQL database pre-populated with the consolidated clinical data schema. The library emulates S3’s file layout to organize unstructured dataset downloads. A lightweight Flask web application replaces interactive dashboards for cohort queries and drilling into associated multimodal records. Python notebooks connect natively to the local database for flexible ad-hoc analysis.

While foregoing autoscaling capabilities, on-premise deployment grants organizations direct control to modify pipelines, incorporate sensitive data, and reduce external network dependencies. The toolkit ensures feature parity while unlocking customizations. The same MySQL structure retains compatibility with predictive models trained in the cloud. Consistent metadata schemas, entity definitions, and configurability guard against lock-in across deployments. By supporting flexible topologies, MINDS balances sovereign data management with scalable cloud analytics.

### 3.5. User Application

The MINDS platform aims to support users across academia, industry, and clinical settings by enabling scalable and secure access to integrated multimodal oncology data. Researchers can leverage MINDS to store, organize, search, and analyze large volumes of heterogeneous data spanning modalities like imaging, sequencing, pathology, and EHRs. For example, a lung cancer researcher may want to analyze treatment response biomarkers across a substantial cohort of lung adenocarcinoma patients. However, gathering sufficient cases poses barriers, as relevant data resides in siloed repositories and trial databases. Each data source may only have a few hundred labeled lung adenocarcinoma cases that meet the desired criteria. Using MINDS, the researcher can easily construct an expansive harmonized analysis cohort. They can perform an SQL query against the aggregated clinical data warehouse to select all lung adenocarcinoma cases across the 95,000+ aggregated case database. This unified view allows for the efficient building of a cohort of over 7000 consolidated lung adenocarcinoma cases—a scale far beyond what any individual source provides. MINDS data processing pipelines will have mapped the clinical data from diverse sources like TCGA, TARGET, GENIE, and clinical trials into a standardized representation aligned with the GDC data model. This harmonizes heterogeneity and structures cohorts for analysis. The researcher can feed this harmonized case ID list into the unstructured data download clients. The tool will automatically retrieve all raw sequencing, imaging, and pathology data objects associated with each case from connected GDC, IDC, and PDC repositories. The researcher now has a turnkey dataset with thousands of consistently structured lung cancer cases annotated with multimodal data. This fuels large-scale integrative experiments to uncover treatment response biomarkers that drive outcomes. By aggregating and standardizing dispersed data into a centralized warehouse, MINDS created an augmented lung cancer cohort at a far larger scale and faster pace than otherwise feasible. This accelerates the discovery process through transformational access to interconnected big data.

## 4. Results and Discussion

This section presents the results of implementing the proposed MINDS architecture for integrated multimodal oncology data management. We demonstrate MINDS’ cohort building and data tracking capabilities and present its advantages over current solutions.

### 4.1. Multimodal Data Consolidation

A fundamental challenge in developing integrated multimodal learning models is assembling the highly heterogeneous and fragmented data from myriad sources into unified datasets at sufficient scale. As shown in Table 1, MINDS directly addresses this by consolidating over 41,000 open-access cancer case profiles spanning diverse research programs into a structured 25.85 MB extract. This aggregated dataset encompasses clinical, molecular, and pathological data elements, providing a multifaceted view of each patient. Compared to petabyte-scale source systems, the extreme compression enables single-node processing and complex SQL analytics that are infeasible on individual repositories. The storage sizes reported for the GDC, PDC, and IDC refer to the total data contained in each repository. However, only a subset of cases in these repositories are open-access and available for research without access restrictions. For example, the GDC contains over 3 petabytes of genomic, imaging, and clinical data overall, but only 17.68 terabytes are associated with open-access cases that can be freely downloaded and analyzed. The 41,499 cases consolidated in MINDS are derived from these open repositories for unencumbered research use.

As shown in Table 2, the consolidated cases represent a comprehensive amalgamation of historical and contemporary research initiatives, vital for maintaining the relevance and accuracy of downstream analytical models in the face of evolving technologies. For example, the 11,315 cases from The Cancer Genome Atlas (TCGA) provide invaluable high-throughput molecular profiling using earlier genomic microarray platforms. In contrast, the 18,004 cases from Foundation Medicine incorporate the latest in contemporary genomic assays, such as next-generation sequencing (NGS) techniques. This strategic blending of data spanning different technological eras—from classic projects like TARGET to modern Foundation Medicine NGS panels—is critical for mitigating chronological biases and batch effects. By integrating this temporally diverse data through MINDS’ heterogeneous integration framework, we proactively inoculate our models against chronological distortions. This approach ensures that the algorithms focus on learning durable, biological patterns that are generalizable across technological shifts rather than transient, platform-specific technical artifacts. Consequently, this temporal synthesis strategy enhances the generalizability of the machine learning models and future-proofs them against inevitable progress in profiling techniques. Access to such a rich and varied dataset is indispensable for training machine learning models, as it provides the large sample sizes necessary for deep learning and helps avoid statistical biases and spurious correlations that often arise from analyzing isolated datasets.

Storage Optimization: By selectively assimilating solely essential clinical, biospecimen, and assay metadata instead of complete image pixel repositories, the MINDS structured ingestion approach reduces storage footprints from original petabyte scales down to a consolidated 25.85 MB extract. This approximate 1000× storage optimization maintains versatile multivariate cohort filtering capabilities across the 41,499 case corpus. Concretely, archiving TCGA, TARGET, and Foundation Medicine oncology profiles requires only MBs—facilitating responsive analytics from single commodity hardware, otherwise impossible at native TB+ scales. MINDS shifts the storage complexity curve through this strategic assimilation to unlock unified exploration. Deferring transfers of raw pixels and nucleic acid sequences until specifically requested for focused analysis prevents excessive upstream overheads. By directly handling initial cohort filtering on structured metadata upstream, MINDS right-sizes infrastructure economics to enable cloud-scale interactivity. Only specifically tailored subsets subsequently retrieve associated imagery. This optimized 2-stage architecture minimizes waste for targeted investigation. We further dissected storage contributions across warehoused clinical, biospecimen, and molecular categories, with cases consuming 10.24MB, followed by genomic variant calls and read groups. This proportional breakdown spotlights metadata categories benefiting from the greatest compression—guiding potential raw assimilation. Measurable storage optimization unlocks interactive analysis otherwise hampered by extreme technical costs, demonstrating quantifiable efficiencies.

Horizontal Scalability: Given that cohort queries constitute read-only analytical workloads, MINDS can scale underlying AWS Redshift compute capacity horizontally by adding managed nodes to meet surging analysis demands transparently. We empirically demonstrate corresponding latency reductions by doubling the cluster nodes, which directly halved runtimes for intensive 8-table cohort investigative queries, proving straightforward scaling. As increasingly complex algorithmic analysis workloads like multimodal federated learning and neural network training expand against MINDS unified corpus, decoupled storage from flexible computing facilitates economic growth, avoiding over-provisioning. This configurable capacity directly fulfills emerging surge requirements without architectural redesign. By empirically plotting reductions in query latencies resulting from MINDS-scaled infrastructure, we substantiate real-world horizontal scalability vital for cloud viability. Similar economical scaling approaches apply when running MINDS datasets through downstream machine learning toolchains. Distributed training frameworks like XGBoost or PyTorch natively support decentralized parallel and vectorized execution pipelines across GPU grids. Maintaining unified data formats ensures interoperability with leading computational platforms.

By harmonizing dispersed data silos into a unified resource, MINDS effectively addresses the primary bottleneck in large-scale multimodal healthcare machine learning model development—a sufficiently large, heterogeneous, and representative dataset for training and validation of models.

### 4.2. Cohort Building

Once aggregated data has been consolidated, tailored cohort extraction is needed to develop optimal machine learning training and test sets. Simple random sampling often fails to provide adequate cohort stratification along key variables. MINDS enables researchers to construct customized cohorts flexibly by querying the unified clinical data using performant SQL.

MINDS implements a flexible end-to-end workflow that allows users to submit analytical cohort queries and receive customized structured or unstructured data extracts. Figure 9 provides an overview of the MINDS system and all the data and query interactions with the user. The process begins with users formulating SQL-based queries that specify criteria to define a cohort of interest. These parameterized queries filter over patient attributes and allow the inclusion of any desired clinical, molecular, or demographic factors. For structured data, the submitted SQL query executes against MINDS’ consolidated EHR database containing harmonized patient profiles. This filtered extraction returns a Pandas data frame containing detailed clinical records for all patients matching the cohort criteria. Alternatively, users can request unstructured data for their defined cohort. In this case, MINDS first extracts a list of unique patient case IDs for those meeting the criteria based on the SQL query parameters. These case IDs are then used to retrieve all associated unstructured medical objects related to those patients from connected repositories. This includes digital pathology slides, medical images like CT/MRI scans, -omics assay files, and other multimodal data assets. This flexible yet automated workflow allows researchers to obtain structured medical records from the EHR or full multimodal datasets matching customized cohorts simply by submitting analytical SQL queries. The tight integration between cohort definition and data extraction enables the on-demand assembly of tailored data corpora for various biomedical applications.

Preliminary experiments demonstrate interactive cohort construction, with simple queries on a single clinical factor completed on average in 3–5 s. Even multidimensional queries joining clinical, molecular, and outcome data across tables are completed within 15 s. This enables rapid, iterative refinement of cohort criteria during model development.

hlQuery Responsiveness: To quantify system performance, we extensively measured SQL query latencies over diverse criteria ranging from simple filters to multidimensional predicates across interconnected data domains. These complex joins emulate realistic exploratory analysis patterns that investigators conduct to uncover relationships within and across data types. Rigorously quantified wall-clock timings over 24,000+ SQL invocations reveal consistent sub-5 s average response times for typical single table queries. More complex multidimensional queries encompassing 8+ tables are completed within 15 s despite traversing metadata for thousands of cases. Minor fluctuations arise based on query types, but critically, sub-15-s overheads enable practically interactive cohort investigation workflows, allowing analysts to rapidly iterate without experiencing disruptions commonplace in legacy repositories. By maximizing fluidity, MINDS facilitates discovering underlying correlations that otherwise remain obscured in fragmented systems.

Researchers have full flexibility to extract customized sets for training algorithms by simply adjusting Boolean logic combining clinical, molecular, or biospecimen factors in the SQL queries. No system constraints are imposed. The ability to interactively construct bespoke cohorts by piping SQL queries directly on consolidated records has several key advantages for multimodal machine learning:MINDS allows researchers to build cohorts tailored to the problem. This prevents sampling biases linked to the availability of pre-defined cohorts.SQL combines and consolidates disparate clinical, molecular, and outcomes data from the entire period of medical treatment. This provides a complete view of each patient.Version IDs uniquely label dataset variants to enable precise tracking of changes during iterative model development. Researchers can pinpoint the exact dataset used to generate each model version.JSON manifests comprehensively log the dataset composition, including the originating queries, data sources, and extraction workflows. This provides full documentation of the data provenance.

### 4.3. Data Standards

The need to integrate data from multiple sources is further pronounced in complex diseases such as cancer, enabling precision medicine and personalized treatments. However, interoperability remains a major challenge in practice despite extensive standards development. Myriad clinical, genomic, imaging, and literature databases use disjoint interfaces, formats, and terminologies—hampering unified analytics. Several domain-agnostic standards have emerged to promote harmonization:Fast Healthcare Interoperability Resources (FHIR): Specifies RESTful APIs, schemas, profiles, and formats for exchanging clinical, genomic, imaging, and other healthcare data. Offers platform-agnostic interconnection.Clinical Data Interchange Standards Consortium (CDISC): Develops data models, terminologies, and protocols focused specifically on clinical research and FDA submissions, including the Study Data Tabulation Model (SDTM) and the Clinical Data Acquisition Standards Harmonization (CDASH).Health Level 7 (HL7): Defines structures and semantics for messaging healthcare data between computer systems, including Clinical Document Architecture (CDA) and Fast Healthcare Interoperable Resources (FHIR) specifications.

However, adopting these standards remains inconsistent, and significant translator development is required to bridge entities. The tight coupling of databases to proprietary representations threatens interoperability. Furthermore, medical ontologies and terminologies play a crucial role in promoting both human and machine-readable shared understanding:Systematized Nomenclature of MEDicine Clinical Terms (SNOMED CT): Provides consistent clinical terminology and codes for EHR. Enables semantic interoperability.National Cancer Institute (NCI) Thesaurus: Models cancer research domain semantics with 33 distinct hierarchies and 54,000 classes/properties. Binds related concepts for knowledge discovery.

Aligning emerging systems like the Multimodal Integration of Oncology Data System (MINDS) with such technologies is vital to avoid isolated silos and enable integrated analytics over clinical and research data. This demands extensive use of their common formats, unique identifiers, controlled vocabularies plus considerable translator development.

### 4.4. Data Tracking and Reproducibility

MINDS further simplifies multimodal analysis by automating the rebuild of full datasets tailored to each cohort. APIs and utilities extract images, -omics, and other unstructured data linked to cohort cases from connected repositories like GDC. Consistent organization and JSON manifest document datasets ready for consumption by machine learning models. To ensure reproducibility, MINDS assigns unique version IDs to cohort datasets. Any changes trigger new versions, enabling precise data tracking to develop different model variants. Comprehensive data provenance from EHR queries to unstructured set regeneration enhances reproducibility in machine learning training pipelines.

### 4.5. Integrated Analytics

Once unified datasets have been constructed, interactive analytics and visualizations are needed to explore cohort characteristics, correlations, and model outputs. MINDS delivers rapid analysis over aggregated multimodal data through integrated dashboards powered by Amazon QuickSight. Optimized cloud data warehousing components like Amazon Redshift enable ad-hoc exploration across thousands of variables without performance lags. QuickSight’s advanced machine learning-driven insights uncover subtle trends and patterns. User-defined charts visualize model performance metrics across various cohorts. Key advantages of integrated analytics include:Rapid hypothesis testing during exploratory analysis to refine cohorts and features.Understanding model performance across cohorts reveals generalization capabilities.Uncovering correlations between clinical factors, assays, and predictions guides feature engineering.Visualizations build trust by providing direct views into model behaviors.

### 4.6. Limitations and Future Improvements

While MINDS has demonstrated significant benefits, there are several areas where the system could be improved. Including controlled data, a local deployment option, and enhanced analytics and visualization capabilities represent exciting directions for future work on MINDS. These improvements would increase the amount of data available in MINDS and enhance its utility for oncology research. Another future extension to this work could be to replicate MINDS on the Google Cloud Platform or Microsoft Azure platform. While there would be specific technical differences across providers, the high-level design focused on abstracted services ensures the seamless prevention of vendor lock-in. Multi-cloud deployments ensure MINDS provides flexible, portable data management capabilities spanning diverse infrastructures. To track the addition, deletions, and modifications to data, webhooks and event notifications can be implemented to achieve more real-time incremental updates. For example, an event trigger could invoke our ingest handler when new data is added to the remote platform. This event-driven approach avoids excessive API polling. Webhooks allow registering listeners to be notified immediately of data changes.

Additionally, though initially focused on cancer data, MINDS’s flexible and modular design makes it well-suited for application across medical specialties. For example, the infrastructure could readily incorporate COVID-19 data types such as clinical outcomes, chest CT scans, and immunological biomarkers from initiatives like the Medical Imaging and Data Resource Center (MIDRC) [59] to accelerate insights. By ingesting such assets via extensions to the automated ETL pipelines and data model while reusing the security, governance, and analytics foundations, MINDS could integrate emerging COVID-19 knowledge. More broadly, maintaining interoperable components enables consolidating distributed data silos across domains to advance data-driven medicine beyond just oncology through unified analytics.

While MINDS demonstrates significant benefits in enabling integrated analytics, some core limitations provide fruitful directions for further enhancement. Given infrastructure barriers, a primary constraint centers on directly ingesting raw clinical imagery and video. However, introducing dimension reduction through learned embeddings holds promise for overcoming such hurdles while preserving semantic representation. Exploring privacy-preserving approaches would also facilitate assimilating regulated data assets beyond public corpora. Additionally, absorbing unstructured physician notes poses non-trivial natural language understanding challenges needing advancement through pre-trained clinical language models. If these addressable constraints are tackled, the potential significance would be immense. MINDS could profoundly transform integrated biomedical investigation paradigms by synergizing heterogeneity and multiplicity across exponentially growing streams. New modalities, data types, and controlled datasets could continually expand the scope. Assimilating free text notes could uncover novel linguistic biomarkers. Exciting enhancements we have highlighted include:Incorporating regulated data through privacy-preserving methodsMigrating imagery via compact embeddingsAbsorbing unstructured notes through advanced NLPExpanding across diseases by reusing consolidation componentsScaling across cloud platforms to prevent vendor lock-in

## 5. Conclusions

The MINDS was designed to address the challenges of integrating and managing large volumes of oncology data from diverse sources. MINDS provides a cost-effective and scalable solution for storing and managing oncology data through its innovative cloud technologies and data mapping techniques. It leverages public datasets to ensure reproducibility and enhance machine learning capabilities while providing a clear pathway for including controlled data in the future. Our results demonstrate that MINDS significantly reduces storage size and associated costs compared to traditional data storage methods. MINDS’ compatibility with public datasets ensures no leaks of controlled data while allowing for reproducibility of results. The system also enhances machine learning capabilities by updating patient information as new data is released from clinical trials, providing transparency and reproducibility.

## Figures and Tables

**Figure 1 sensors-24-01634-f001:**
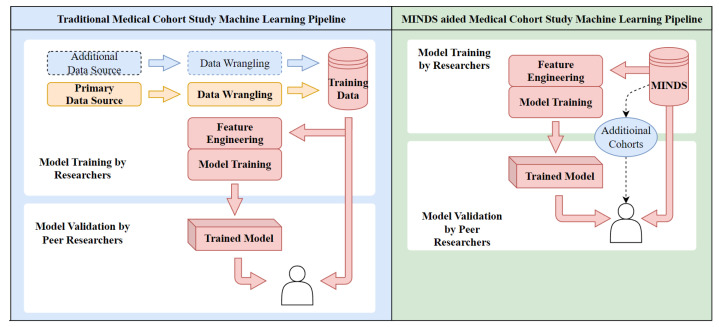
In the first stage, data is obtained from a primary source that may consist of patients who comprise the bulk of your study. Still, access to additional data sources may be needed to augment the original dataset, which may be restricted, or a minimal amount of such data that aligns with the original cohort may be present, limiting the ability to develop comprehensive models. Once collected, both sets undergo their respective data wrangling. It can be time-consuming and resource-intensive when new data sources are introduced due to different data cataloging and formatting pipelines. Next, the model development phase consists of feature engineering and model training. However, once developed, the model’s effectiveness is limited by the scope of the original dataset. Validating and expanding existing models with new data sources requires extensive data wrangling and matching the data pipelines of the original researcher, leading to longer validation times and decreased productivity. Consequently, the broader applicability of these models remains unexplored mainly, limiting the potential for advances in understanding complex diseases and developing effective treatments. In contrast, MINDS integrates multimodal data into a unified platform, and researchers can employ early-stage data fusion techniques that can harness the rich potential of correlated multimodal data to improve inference and decision-making. For instance, in the context of medical applications like cancer research, the integration of MR, X-ray, and ultrasound imaging data with different modalities of data acquired not from scans, such as histopathology slides, can yield more accurate and comprehensive insights into patient conditions compared to relying on any one modality alone.

**Figure 2 sensors-24-01634-f002:**
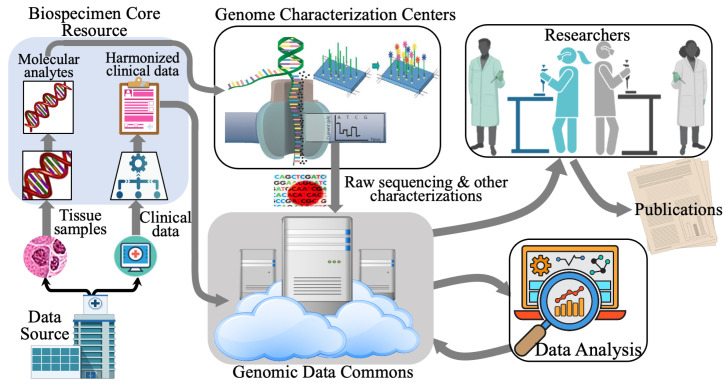
Genome Characterization Pipeline is illustrated as an example of data characterization. Data source sites collect tumor tissue samples and normal tissue from participating patients. Biospecimen Core Resource (BCR) collects and processes the tissue samples and collects, harmonizes, and curates clinical data. Genome Characterization Centers (GCCs) generate data such as whole genome sequencing, total RNA and microRNA sequencing, methylation arrays, and single-cell sequencing from the tissue samples received from the BCR. At the Genomic Data Analysis stage, the raw data from the previous stage is transformed into meaningful biological information. Data generated by the pipeline are made available via the GDC for use by researchers worldwide. Center for Cancer Genomics (CCG) Genome Characterization Pipeline was originally published by the National Cancer Institute [21].

**Figure 3 sensors-24-01634-f003:**
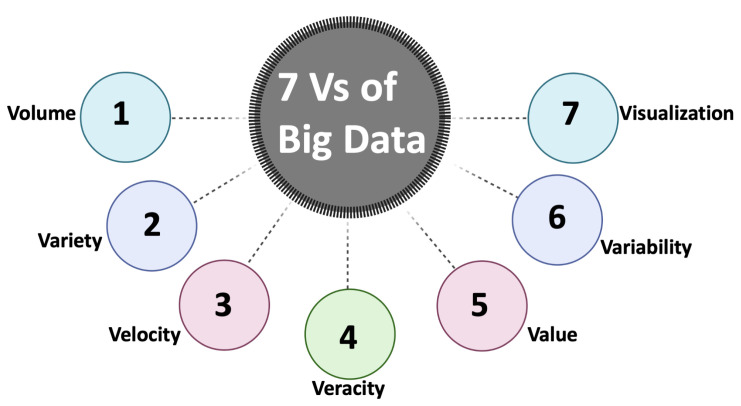
The 7 Vs of Big Data: Volume relates to the data size, with more data, the models can learn more and perform better. Variety refers to the data types we deal with, each data type presents unique challenges and opportunities. Velocity considers the speed at which the data is accumulated, the learning models need to remain current and adaptable. Veracity concerns the quality and integrity of the data, data must be credible and high-quality. Value focuses on the utility and benefits of the data. Variability pertains to the data volatility that changes in both temporal and spatial domains. Visualization depicts insights through visual representations and illustrations [40].

**Figure 4 sensors-24-01634-f004:**
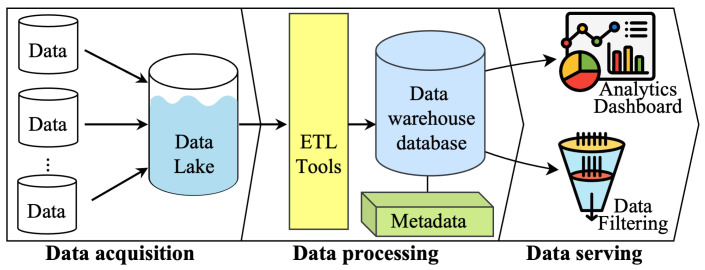
MINDS architecture implements a 3-stage pipeline designed to optimize data aggregation, data preparation, and data serving of multimodal datasets. Stage 1 comprises data acquisition and involves acquiring structured and semi-structured data from sources like GDC, including clinical records and biospecimen metadata. These are gathered, normalized, and securely stored in cloud object storage. Stage 2 consists of data processing. The raw data is processed by extract, transform, load (ETL) tools cataloging into data lakes, transforming into structured relational formats, and loading into optimized data warehouses, generating analysis-ready clinical data. Stage 3 consists of data serving. The clinical data is served directly to researchers for preliminary exploration and visualization. They can also build patient cohorts by querying the selection criteria, and MINDS will pull corresponding unstructured data like images from connected repositories, e.g., IDC.

**Figure 5 sensors-24-01634-f005:**
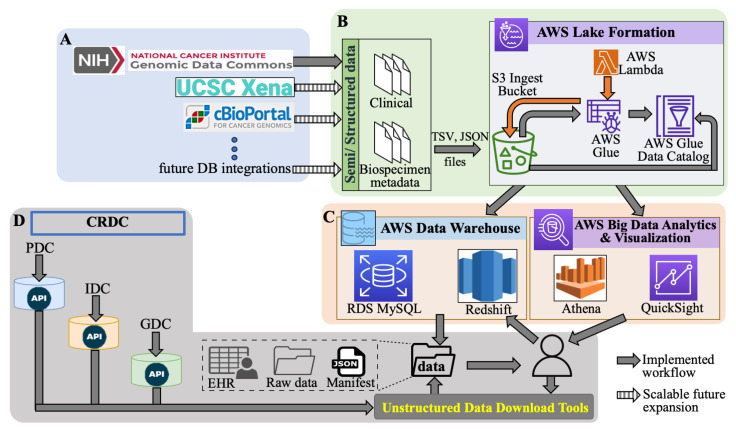
Overview of the MINDS architecture implemented on AWS. (**A**) Data from multiple oncology sources is acquired. The pipeline for structured data is currently configured with GDC, with the ability to integrate other platforms, such as the University of California Santa Cruz Xena and cBIO portals. (**B**) The structured data from the source is acquired in an AWS Lake where multiple components such as S3 Bucket, Glue, and Lambda catalog and process the data. (**C**) Next, the Data Warehouse uses RDS and Redshift for structured data warehousing in the form of relational schema. The cataloged data is available to Athena and Quicksight for analytics and visualization. (**D**) The users can directly query the structured data for visualization. All unstructured data download pipelines using the Data Commons APIs from Cancer Research Data Commons (CRDC) are also shown. Using SQL queries, users can request all modalities data associated with the cohort. Resultantly, all the data from PDC, GDC, and IDC are pulled together, harmonized, formatted, and presented to the user ready to use for machine learning pre-processing.

**Figure 6 sensors-24-01634-f006:**
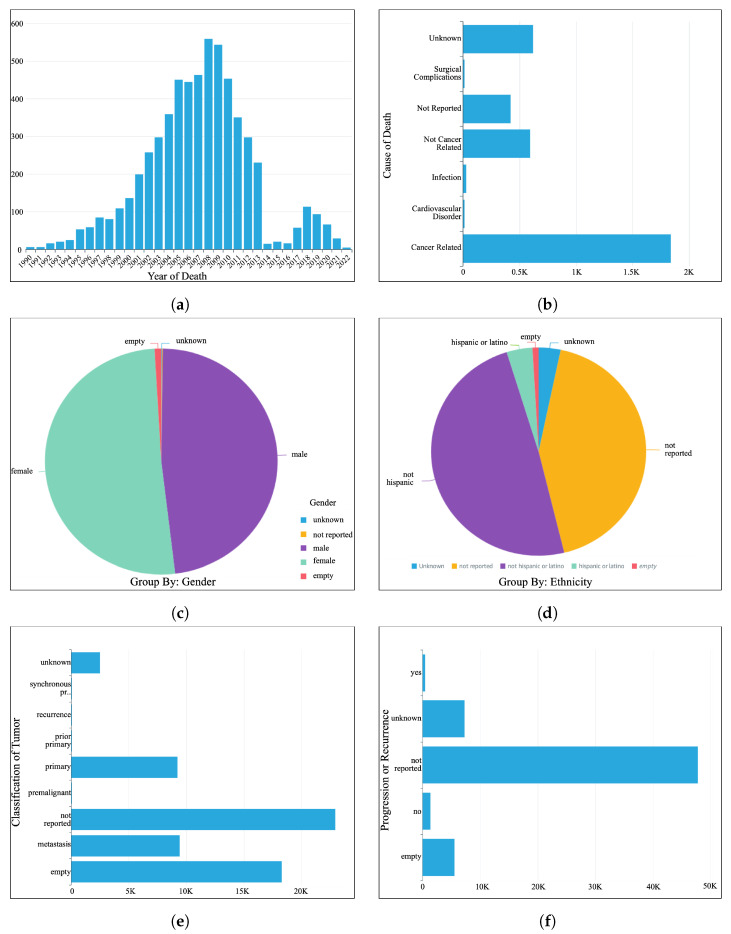
Quicksight analytics and visualization generated using clinical data from MINDS, filtered based on the condition mentioned in each sub-figure. Showcasing data mining and hypothesis generation capacity based on querying MINDS’ consolidated case data and deriving tangible trends. The presented visualizations offer glimpses into the extensive cohort analytics and visualization capacities, where MINDS aims to accelerate discoveries by surfacing multidimensional correlations. (**a**) Count of Records by the Year of Death. (**b**) Count of Records by the Cause of Death. (**c**) Count of Records by Gender. (**d**) Count of Records by Ethnicity. (**e**) Count of Records by Classification of Tumor. (**f**) Count of Records by Progression or Recurrence.

**Figure 7 sensors-24-01634-f007:**
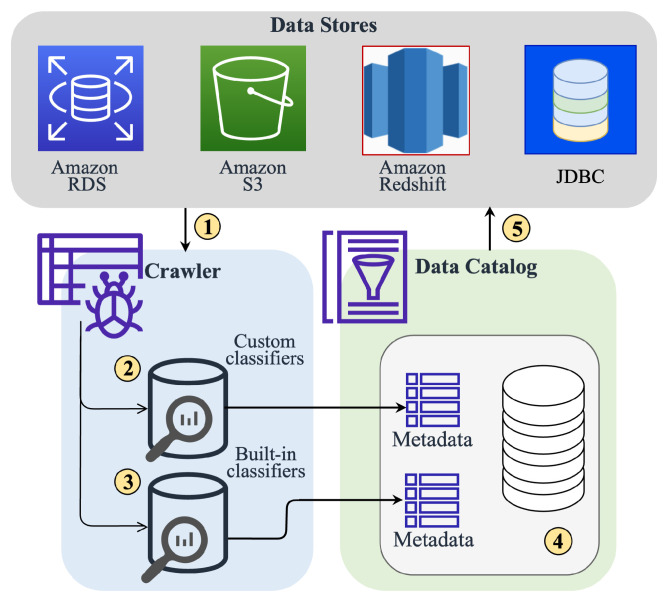
The AWS Glue crawler automates ETL in MINDS through a 5-step workflow. (1) Establish secure database connections. (2) Apply custom classifiers to catalog raw data. (3) Transform data using built-in classifiers. (4) Merge classifier outputs into unified databases. (5) Upload the final catalog to processed data stores. The proposed workflow extracts, standardizes, and structures heterogeneous multimodal data from diverse sources to enable advanced analytics applications.

**Figure 8 sensors-24-01634-f008:**
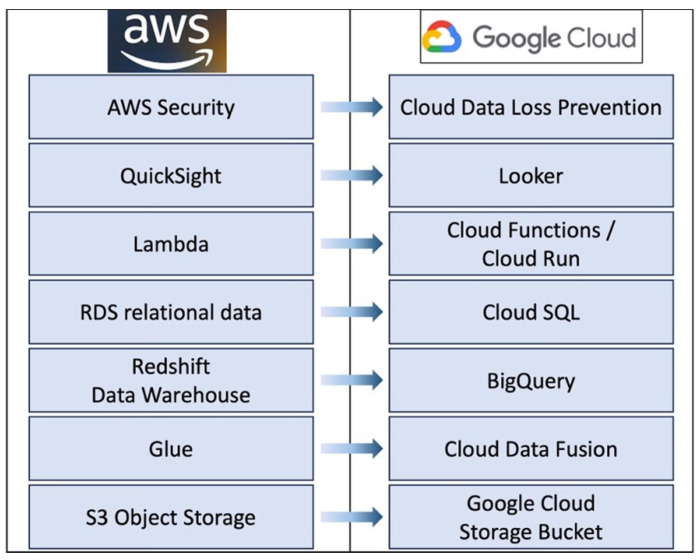
Demonstrating the feasibility of deploying MINDS across cloud platforms, this diagram shows the mapping of key AWS services leveraged in the current implementation to their corresponding managed offerings on Google Cloud Platform (GCP). By abstracting underlying infrastructure into modular cloud services with standardized programmatic interfaces, MINDS aims for platform agnosticism without vendor lock-in. While technical considerations around service limits, access controls, and performance tuning differ across providers, the high-level architecture and methodology remain consistent. Through this interoperability, MINDS can ingest, process, analyze, and serve integrated multimodal datasets spanning storage systems, data pipelines, warehouses, and analytics products from multiple public cloud platforms.

**Figure 9 sensors-24-01634-f009:**
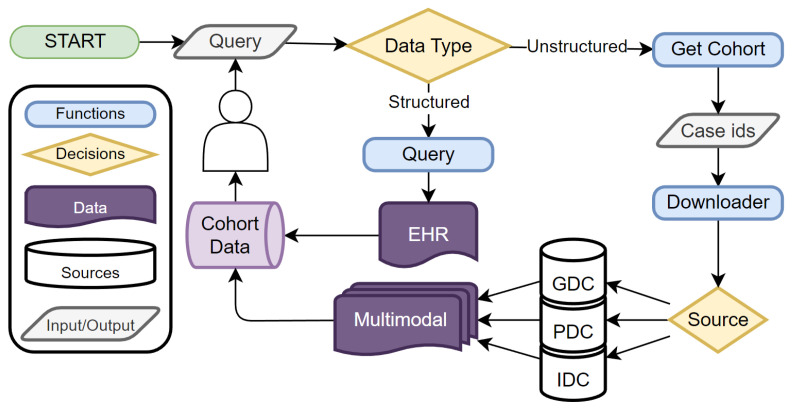
Overview of the workflow in MINDS, starting from user query generation through returning the cohort data, structured and unstructured. The system starts with a user submitting an analytical query specifying cohort criteria. If the user requests structured data, the query is sent to a function that executes it against the consolidated EHR and clinical databases, returning a Pandas data frame containing matching patient records. Alternatively, if the user requests unstructured data for the cohort, the query is sent to another function that extracts a list of unique case IDs for patients meeting the criteria. This case list is then used to retrieve all associated unstructured data objects like medical images, genomic sequences, and pathology slides for those patients from connected repositories, including GDC, PDC, and IDC. The cohort-specific unstructured data extract is returned to the user for further analysis.

**Table 1 sensors-24-01634-t001:** Comparison of the storage size of structured clinical and metadata extracted in MINDS versus complete unstructured data holdings in the GDC, IDC, and PDC repositories. MINDS only consolidates structured information like patient records and biospecimen data. Raw unstructured data, including medical images, genomic sequences, and digital pathology slides, remain hosted separately in their respective source platforms. The storage sizes reflect this distinction between structured extracts in MINDS and total unstructured data in the commons. The comparison illustrates the extreme compression of MINDS’ structured approach versus petabyte-scale repositories containing all raw imagery and assay data.

Data Source	Storage Size	# of Cases
MINDS	25.85 MB	41,499
PDC	36 TB	3081
GDC	3.78 PB (17.68 TB public)	86,962
IDC	40.96 TB	63,788

**Table 2 sensors-24-01634-t002:** Distribution of the number (#) of cases by programs from GDC open cases present in MINDS.

Program	# of Cases
Foundation Medicine (FM)	18,004
The Cancer Genome Atlas (TCGA)	11,315
Therapeutically Applicable Research to Generate Effective Treatments (TARGET)	6542
Clinical Proteomic Tumor Analysis Consortium (CPTAC)	1526
Multiple Myeloma Research Foundation (MMRF)	995
BEATAML1.0	756
NCI Center for Cancer Research (NCICCR)	481
REBC	440
Cancer Genome Characterization Initiatives (CGCI)	371
Count Me In (CMI)	296
Human Cancer Model Initiative (HCMI)	228
West Coast Prostrate Cancer Dream Team (WCDT)	99
Applied Proteogenomics OrganizationaL Learning and Outcomes (APOLLO)	87
EXCEPTIONAL RESPONDERS	84
Oregon Health and Science University (OHSU)	80
The Molecular Profiling to Predict Response to Treatment (MP2PRT)	52
Environment And Genetics in Lung Cancer Etiology (EAGLE)	50
ORGANOID	49
Clinical Trials Sequencing Project (CTSP)	44

## Data Availability

The data integrated into MINDS is sourced from publicly available datasets in the Genomic Data Commons (https://gdc.cancer.gov, accessed on 12 January 2024), Proteomics Data Commons (https://proteomics.cancer.gov, accessed on 12 January 2024), Imaging Data Commons (https://imaging.cancer.gov, accessed on 12 January 2024) and cBioPortal for Cancer Genomics (https://www.cbioportal.org, accessed on 12 January 2024). Only metadata and identifiers are ingested into MINDS—the datasets remain hosted in their respective repositories. Analyses are performed by querying these sources through their public APIs and computational workbenches. The code implementation for the MINDS platform is available at https://github.com/lab-rasool/MINDS, accessed on 11 January 2024.

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
