# Peer review of "Building Flexible, Scalable, and Machine Learning-Ready Multimodal Oncology Datasets"

_sensors, 2024, doi:10.3390/s24051634_

Round 1

Reviewer 1 Report (Previous Reviewer 3)

Comments and Suggestions for Authors

The authors have responded very well to my remarks. I have no further comments.

Author Response

Reviewer 2 Report (New Reviewer)

Comments and Suggestions for Authors

Reviewer found that the authors have done an interesting study. There are a few comments below.

1. Line 316: What specific Big Data processing methods were used in your research, and what results did they bring?

2. Fig. 2: What specific technologies and tools are used in each of the three stages of the MINDS pipeline for processing and aggregating medical data?

3. Line 461: What specific methods and tools were used to collect and harmonize data from the GDC portal?

4. Line 640: What aspects of AWS data security have been used to protect the MINDS data warehouse, and what security measures have been implemented to protect data at rest and in transit?

5. Line 723: What specific research initiatives have been included in the consolidated cases, and what practical contribution can they make to the development of analytical models?

6. Line 727: How does combining historical and modern cohorts of data help protect against batch effects and chronological distortions in machine learning models, and what advantages can this bring to identify hidden patterns?

7. Line 762: What specific methods and technologies have been used to ensure the rapid execution of queries on multidimensional data and what aspects of the system architecture allow for high performance in the formation of cohorts?

8. What measures have been taken to ensure data security when executing custom sets for learning algorithms, especially considering the possibility of making changes to the logical sequence of SQL queries and the potential risks associated with this?

9. What prospects do you have for expanding the MINDS data warehouse beyond the GDC and what challenges do you see when integrating new data sources?

10. What does this study add to the subject area compared to other published material?

11. Conclusions lack of limitation, significance, and potential development of this method.

Author Response

Reviewer 3 Report (New Reviewer)

Comments and Suggestions for Authors

I am happy to see innovativeness in the idea, but even then, the proof of implementation, time, and space complexity analysis are missing. Moreover, this seems to be just an idea of deliverability; performance has not happened, and no such proof was provided. The idea is more appreciable, but its supportiveness to the community is not proven in terms of experimental results, or the ideology is not delivered in a convincing manner of innovativeness. It speaks about a general methodology of storing multiple datasets in Amazon web services.

I hope it needs some proof of time and space complexity and some implementation results as a case study.

Author Response

Reviewer 4 Report (New Reviewer)

Comments and Suggestions for Authors

1. The authors should make the introduction and review of literature much shorter and crisper for the better understanding of readers.

2. What do the authors mean by the term “machine learning-ready” in the manuscript? Does it mean that the authors will provide some pre-processed data ready for application in machine learning? If not so, any other preferable term should be used instead.

3. The method section should only contain the details of the work done by the authors not detailed review analysis of each of the steps. The authors should write the methodology accordingly.

4. The authors should provide short description about how the users can use the MINDS for their work purpose.

Round 2

Reviewer 4 Report (New Reviewer)

Comments and Suggestions for Authors

The revised manuscript may be accepted for publication.

This manuscript is a resubmission of an earlier submission. The following is a list of the peer review reports and author responses from that submission.

Round 1

Reviewer 1 Report

Comments and Suggestions for Authors

Reject and do not reconsider, I have some major concerns:

1: In this study, the proposed framework is totally unreliable.

2: There is no novelty in this work

3: The data collection leads towards biasness in this research.

4-Abstract needs to be enhanced by adding performance measure scores. Also, please highlight your findings.

5. Please highlight your contributions in the introduction section.

6- In related work, the author did not mention the gap they identified from previous studies.

7: Enhance the quality of the figures in the manuscript.

8-The manuscript is missing the future work section.

9-Please proofread the manuscript. I found many typos and grammatical errors. 

10:There is a need for extensive English revision.

Comments on the Quality of English Language

 Extensive editing of English language required

Reviewer 2 Report

Comments and Suggestions for Authors

I have no major concerns to raise, but rather a few points that I would be  keen to learn more about:

- the platform and functionalities are described  in terms of its  use in a particular cloud provider; how would the authors envisage its deployment and usage in other could providers;

- concerning teh update of the information, it is described to be triggered by the arrival of new data to S3... but what prompts that arrival? An automated process?

- As for the comparison in Table 1, does it account for all  kinds of information data (eg images) 

Reviewer 3 Report

Comments and Suggestions for Authors

The MINDS framework is certainly very useful but the paper needs to be worked on more. MINDS needs to be put in a wider context than just its connection with a number of NIH and NCI hosted databases.

Comments:

- wouldn't this paper better suit an MDPI journal like "Data"?

- The introduction should describe how MINDS compares to or fits in existing initiatives and papers such as (but not limited to) PMIDs 37767106, 34870187, 36220072 or even older ones such as 17911733. In fact, the discussion section states to "present its advantages over current solutions", but then fails to do so. The idea of an integrated oncology data system is certainly not new.

- Figure 5 has minimal introduction in the text. What insights can be gained from these graphs? What happened with the "year of death" between 2014 and 2017? Is this a data quality issue?

- For data integration, interoperability is key. FAIR is mentioned briefly, but I would also expect more discussion about standards, ontologies, FHIR HL7, etc.

- Discussion: MINDS is focused on oncology, but wouldn't this system be useful in other disease areas as well?

Minor comments:

- please sort the abbreviation table

- page 17: EHR was already introduced on page 2
